# Integrated Mycotoxin Management System in the Feed Supply Chain: Innovative Approaches

**DOI:** 10.3390/toxins13080572

**Published:** 2021-08-16

**Authors:** Francesca Fumagalli, Matteo Ottoboni, Luciano Pinotti, Federica Cheli

**Affiliations:** 1Department of Health, Animal Science and Food Safety, “Carlo Cantoni” University of Milan, 20134 Milan, Italy; matteo.ottoboni@unimi.it (M.O.); luciano.pinotti@unimi.it (L.P.); federica.cheli@unimi.it (F.C.); 2CRC I-WE (Coordinating Research Centre: Innovation for Well-Being and Environment), University of Milan, 20134 Milan, Italy

**Keywords:** feed mycotoxins, modified mycotoxins, emerging mycotoxins, mycotoxin co-occurrence, feed safety, HACCP, integrated mycotoxin management system, sustainability, biotechnologies, nanotechnologies

## Abstract

Exposure to mycotoxins is a worldwide concern as their occurrence is unavoidable and varies among geographical regions. Mycotoxins can affect the performance and quality of livestock production and act as carriers putting human health at risk. Feed can be contaminated by various fungal species, and mycotoxins co-occurrence, and modified and emerging mycotoxins are at the centre of modern mycotoxin research. Preventing mould and mycotoxin contamination is almost impossible; it is necessary for producers to implement a comprehensive mycotoxin management program to moderate these risks along the animal feed supply chain in an HACCP perspective. The objective of this paper is to suggest an innovative integrated system for handling mycotoxins in the feed chain, with an emphasis on novel strategies for mycotoxin control. Specific and selected technologies, such as nanotechnologies, and management protocols are reported as promising and sustainable options for implementing mycotoxins control, prevention, and management. Further research should be concentrated on methods to determine multi-contaminated samples, and emerging and modified mycotoxins.

## 1. Introduction

Animal feeds have an important role in the world food industry, empowering the safe production of animal-origin food across the globe. The feed industry is an integrated part of the food chain, and it generates income and economic sustainability. Feed safety is a precondition for food safety and human health [1] as well as a requirement for animal welfare and health; it has been acknowledged as a shared value and responsibility among all production steps.

Mycotoxins are toxic compounds formed by the metabolism of specific fungi that affect crops and contaminate commodities consumed by humans and animals. Fungal growth rely on favourable environmental conditions [2,3,4]. Exposure to mycotoxins is a worldwide concern [5], and their occurrence is unavoidable and varies among geographical areas [6]. With the globalization of feed ingredients, trade, and climate changes, the occurrence of mycotoxins becomes gradually difficult to predict [7,8]. Mycotoxins have been declared a high priority by the Food and Agriculture Organization of the United Nations (FAO) and by the World Health Organization (WHO) due to their toxicological impact on human and animal health. This has led to legislative limits for mycotoxins in about 100 countries, with regional harmonisation for the European Union (EU), Australia and New Zealand (AU & NZ), the Gulf Cooperation Council (GCC), and Mercado Común del Sur (MERCORSUR; Argentina, Brazil, Paraguay, Uruguay, and Venezuela). In the EU, maximum levels in feed are enforced for Aflatoxins (AFs), Deoxynivalenol (DON), Fumonisins (FUM), Ochratoxins (OT), Zearalenone (ZEA), and T-2 and HT-2 toxins. The legal limits have been stipulated for AFs, while for other toxins, there are national and international recommendations [9,10,11,12,13]. Another topic of concern is the co-occurrence of mycotoxins [14]. Feed may be contaminated by several fungal species and mycotoxins at the same time, and the toxicological effects can be different according to the type of mycotoxin interaction: less than additive, additive, synergistic, enhanced, or antagonistic [15,16,17]. For these reasons, mycotoxin concentrations in feed should be continuously monitored to support risk assessment.

Although a number of methods can be used, it is almost impossible to prevent mould and mycotoxin contamination. It is widely recognised that mycotoxins are classified as chemical or biological hazards; therefore, effective quality control methods, such as HACCP and GMP, should be implemented including mycotoxin control [18,19,20]. It is necessary for producers to enforce a comprehensive mycotoxin management program to minimize these risks along the whole animal feed supply chain [18].

Mycotoxin management is systemic and includes all stages of the feed supply chain, starting from the production of raw materials to the feeding in the farm: crop phase (pre-harvest and harvest), transportation, storage, feed mill operations, and livestock production.

There are many publications on mycotoxins in animal feed, but few relate to an integrated system for their management. Moreover, the existing ones do not include some innovative methods for mycotoxin management such as biotechnology and nanotechnologies. The aim of this paper is to propose an updated integrated system for managing mycotoxins in the feed production chain, with a focus on novel strategies for mycotoxin control.

## 2. Mycotoxin Occurrence

Recently, surveys have been performed to assess the worldwide incidence of mycotoxin contamination in feed and raw feed materials, mainly grains and grain co-products (corn gluten meal, bran, and dried distillers’ grains) as well as other feed ingredients, although these are to a minor extent (e.g., soybean meal, cotton seed, sorghum, peanut, copra, cassava, etc.). These surveys state that AFs, DON, FUM, ochratoxin A (OTA), T-2 toxin, and ZEN are the principal contaminating mycotoxins in feed [21,22,23,24]. The results of the mycotoxin surveys highlighted two important issues of great concern for feed safety: mycotoxin co-occurrence, and modified and emerging mycotoxins [25].

The world mycotoxin survey [5] has been recently published, in which the European situation in 2020 has been analysed in comparison with the previous year: risk in Europe is high to severe. The most ubiquitous mycotoxin is DON, followed by ZEN and FUM. DON is the main hazard for livestock, with 70% of corn samples positive for this mycotoxin. Cereals were also a concern: DON reached a concentration of 11,875 ppb. ZEN increased its average contamination in corn to 171 ppb. Regarding AFs, their contamination is more prevalent in southern Europe, where they reached up to 20% of the positive cereals samples. This data have also been associated with climate change. For instance, Fusarium incidence was low or absent in the most southern regions of Italy and Spain until a few years ago; however, during the last years several northern regions of Italy, Spain, and Portugal as well as some southern regions of France and the Balkan Peninsula, Fusarium graminearum increased its occurrence in cereals at maturity, together with DON, inducing a high occurrence of this mycotoxin also in southern Europe. Regarding AFs, their contamination is growing in the Mediterranean area, where extreme changes in temperature, CO_2_ levels, and rainfall patterns in combination with high heat and drought seems to compromise host plant resistance and therefor facilitates *A. flavus* infection.

To obtain significant data regarding mycotoxin occurrence in feed and food, sampling and analysis are the critical points. The Commission Regulations [10,26] setting down the sampling and analysis methods for the official control of the levels of mycotoxins in feed and foodstuffs are in force.

### 2.1. Mycotoxin Co-Occurrence

The probability of finding only one mycotoxin contaminating raw materials or feed is extremely low. Worldwide, the incidence of co-contamination is high. The global monitoring reported that 72% of the samples of feed and raw materials were contaminated with more than one mycotoxin [27]. The same authors [28] detected 83 samples of feed and raw materials contaminated with 7 to 69 mycotoxins per sample, having analysed 169 different compounds.

The occurrence of mycotoxin co-contamination in Europe tends to follow the same pattern. Several studies have revealed the simultaneous presence of mycotoxin co-contamination in samples from European countries, finding a high percentage of feed samples contaminated with trichothecenes (DON, acetyldeoxynivalenol (AcDon), T2, and HT2) and FUM, as well as with ZEN [29,30,31,32,33,34,35,36]. In Germany, [37] maize was found simultaneously contaminated with 14 Fusarium mycotoxins, such as DON and its acetylated forms, ZEN, Moniliformin (MON), Beauvericin (BEA), Nivalenol (NIV), Eniantins (ENNs), FBs, and HT-2 Toxin. Recently, data have been published regarding fodder mycotoxin co-contamination, which showed all silage samples positive for at least one mycotoxin, and 61% of samples contained five or more mycotoxins simultaneously. According to [5], an average of 30 mycotoxins and their metabolites per sample were found and 87% of the samples have 10 or more mycotoxins or metabolites.

The most frequently detected toxins were DON, NIV, ZEA, enniatins, and BEA, although the levels of these toxins were relatively low [38]. Co-contamination is a great concern, as it may exert adverse effects on animals due to the additive/synergistic interactions of the mycotoxins, the complexity of which varies according to the animal species, the level and type of mycotoxin contamination, the toxicity of the compound ingested, body weight, age and animal physiological condition, compound action mechanism, the presence of other mycotoxins, and the length of exposure. In general, in most cases, there are additive or synergic effects [39,40]. Many authors have highlighted this additivity, synergy, or enhancement. Although most results reveal the additive or synergic effects of mycotoxins, it should be noted that antagonistic effects could also be seen [41,42,43]. The co-occurrence between regulated, modified, and emerging toxins and their interactions are still little known.

### 2.2. Modified Mycotoxins

The European Food Safety Authority (EFSA) refers to “modified mycotoxins” as all forms that have been structurally modified in relation to their “parental compound” or the free mycotoxins [44,45]. Plants and certain microorganisms, such as yeasts, filamentous fungi, and bacteria, are capable of transforming mycotoxins into conjugated forms (biologically modified mycotoxins), reducing their toxicity [46]. In fact, plant metabolites have been identified for DON, NIV, fusarenon-X, T-2 toxin, HT-2 toxin, ZEN, OTA, destruxins, and fusaric acid, while modified fumonisins have been detected in cereal commodities, such as corn, wheat, and barley. Toxicological data on modified mycotoxins, including those of processing origin (chemically modified mycotoxins), are still limited [47,48]. However, recent advances in modified mycotoxin occurrence and toxicity have suggested that mycotoxins conjugates have a reduced toxicity potential due to the lower absorption in the gastrointestinal tract [6]. These modified mycotoxins differ in their structure, solubility, polarity, and molecular mass; furthermore, they can be formed during the processing of foods from contaminated raw materials and can be reconverted to the original toxin during the human and animal metabolism [49,50,51,52]. Free mycotoxins co-occur with modified mycotoxins [49,50], and the modified mycotoxin concentration exceeds the level of free form. The possibility of modified mycotoxin conversion to its free form may involve risks for human and animal health. The conversion of modified to free form can result in increased bioavailability of mycotoxin [53,54]. It is necessary to set up and validate affordable methods for the detection of modified mycotoxin as well as to study their stability during the processing of feeds, their outcome in the animal digestive system, and their toxicokinetic and toxicodynamic properties [55]. In addition, the knowledge of their formation process and their structure and molecular mass may resolve analytical and technological gaps [56,57].

### 2.3. Emerging Mycotoxins

Emerging mycotoxins became a major issue due to their high occurrence in cereals, feed, and food commodities [58,59,60]. They are lesser-known or newer forms of mycotoxins that, by definition, are neither routinely determined nor legislatively regulated. The most prevalent emerging mycotoxins are *Fusarium* toxins, such as ENNs, BEA, MON, fusaproliferin (FP), fusidic acid (FA), culmorin (CUL), and butenolide (BUT) [61,62]. The presence of emerging mycotoxins in feed and food commodities, such as *Aspergillus* toxins (sterigmatocystin (STE) and emodin (EMO)), *Penicillium* toxins (mycophenolic acid (MPA)), *Alternaria* toxins (alternariol (AOH), monomethyl alternariol ether (AME), altenuene (ALT), altertoxin (ATX), and tenuazonic acid (TeA)), ergot alkaloids, and citrinin, is equally common [58,61,62]. Research indicates that these emerging toxins are rapidly becoming prevalent co-contaminants in feed and food such as grains (corn, wheat, barley, etc.) showing greater occurrence when other *Fusarium* mycotoxins are present. In an extensive review on co-occurrence of regulated, modified, and emerging mycotoxins in finished feed and maize, emerging mycotoxins, such as ENNs, MON, and BEA, were found to be ubiquitous in analysed samples [48]. According to [5], the raw materials most frequently presenting emerging toxins are maize and animal compound feed. In particular, corn presented 93% of MON and 83% of Aurofusarium, while the finished feed presented 97% and 93% for Bauvericin, and Enniantin B and B1.

However, these data have to be considered with caution. Indeed, although thanks to certain modern analytical methods we are able to detect hundreds of “new” different fungal metabolites in a variety of food and feed samples, we have to consider that many of these compounds are irrelevant in terms of food and feed safety [62]. Their limited risk, however, can change in the future: climate change, commodities origin, and processing as well as several others environmental factors can alter both toxicity and occurrence of these compounds. Thus, for an adequate risk assessment and in order to prevent future food and feed safety crisis, it is also important to start collecting information (occurrence, acute vs. chronic toxicity, distribution, commodities, etc.) about these fungal metabolites.

## 3. Mycotoxins and Climate Change

Mould growth and subsequent mycotoxin production is highly dependent on environmental or extrinsic factors, such as relative humidity, temperature, and oxygen/CO₂ [63,64]. However, intrinsic factors such as ingredient composition, pH, grain moisture, and water activity, play important roles in the potential for mould growth and mycotoxin production [63,64].

Climate change has an impact on the occurrence of feed/food safety hazards along the whole agri-food chain from farm to fork. There is a significant interest in understanding the climate change-related abiotic factor impact (increased temperature, elevated CO₂, and extremes in water availability) on the relative risks of mycotoxin contamination and influences on feed/food safety and security [65,66]. Extreme climate conditions such as floods and droughts, which have not commonly occurred in the past, may be factors in crop contamination by various species of toxigenic fungi and related mycotoxins [67,68]. Fungal plant pathogens are expected to move globally and to change the diversity of diseases and pests invading essential crops with economic and social expenses. The European Food Safety Authority (EFSA) has analysed the potential impact of climate change in Europe and has advised that these effects will be regional and negative, or advantageous depending on geographical region [69]. The effects of climate change on cereals are considerable and detrimental as ripening in southern and central Europe will occur much earlier than currently. This will affect pests and diseases, reducing yields, increasing mycotoxin contamination, and generating a severe impact on food security and safety in different continents. Based on currently available data, CO₂ atmospheric concentrations are expected to double or triple in the next 25–50 years. Different regions in Europe will be impacted by temperature increases from 2 to 5 °C with elevated CO₂ and drought episodes. An analogous scenario was expected in other areas of the world, especially Asia, Central, and South America—the global wheat, cotton, maize, and soya beans producers [70]. The climate change impact is correlated with both nutritional/quality losses and toxin exposure, particularly in Low and Middle Income Countries (LMICs). Environmental stress has been revealed to have significant implications for mycotoxin production. Elevated CO₂ causes a consistent effect on the mycotoxin content [68]; climate change has radically influenced the global agricultural sector through restrictions in water, vegetative land, and temperature elevation that increase humidity. The high humidity raises moisture levels, enabling fungal growth and mycotoxin formation [71].

## 4. Mycotoxin Risk in the Feed Supply Chain: Need for a Management Plan

Commodities can become contaminated with mycotoxins anytime in the production cycle, i.e., at each stage from the field through harvesting, processing, storage, and transportation [72,73]. A representation of the feed chain is shown in Figure 1. Each of these phases are examined in Section 5 and Section 6.

The feed chain is complex and articulated; the same applies to mycotoxin contamination. The demand for feed will increase by 2050 in support of animal product requests; in view of this growth, feed safety will become fundamental. There are steps in the feed process that can be updated through the experimentation of new technologies, some of which are designed to manage mycotoxin contamination.

One possible approach for managing mycotoxin risks in the feed supply chain is the use of an integrated system [74] (Figure 2).

An integrated system includes technical aspects such as fixing regulatory limits, programming a precise monitoring and control of cultivation, and production phases. In addition, it proposes solutions to non-conformities that may take place and, above all, for widespread training of all operators in the feed chain. An integrated system is therefore preventative, planning how to act in the case of system anomalies. It is a plan to avoid arriving unprepared in the case of contamination.

An integrated system is based on the synchronised use of prevention and control implements such as Good Agricultural Practices (GAP), Good Manufacturing Practices (GMP), Good Hygienic Practices (GHP), quality control, and Hazard Analysis and Critical Control Point (HACCP) at all stages of production from the field to the final consumer. The phases of an integrated mycotoxin management system as proposed by FAO in 1995 [74] are reported in Table 1. Although most of these actions (legal limits, control systems, alert systems, etc.) have been activated in different world regions by single countries or market/areas (e.g., EU), world harmonisation exists.

### HACCP for Feed Mycotoxin Contamination

In an integrated management system, prevention is key; the risks related to mycotoxin hazards should be minimized at each production phase. Mycotoxin contamination is best dealt with in the pre-harvest phase, but when contamination occurs, the related dangers can be handled through post-harvest techniques, applying the corrective actions reported in the HACCP plan.

The HACCP system is designed to decrease the risk of feed safety exposure by identifying the hazards and by monitoring controls [74,75]; it provides a scientific quality control methodology. In addition, HACCP system can be designed and used in combination with other quality systems. The most crucial source of mycotoxin intake can be found in agro-commodities (FAO/WHO 2014); for this reason, an effective preventative strategy could be represented by an approach spanning the entirety of the commodity supply chain. The application of an HACCP system aimed at improving food safety, from the fields to farm animals, can control mycotoxin contamination of raw material [76,77]. To ensure that the product has acceptable mycotoxin levels, an integrated HACCP approach in the pre-harvest stage can be used. Effective integrated management programmes cover agro-products mycotoxin prevention/detoxification as well as routine surveillance, updating national and international regulatory measures, information, education, and communication activities.

The key points for an effective HACCP plan are hazard identification and analysis and the record keeping procedures [78,79,80]. The text “Manual on the application of the HACCP system in mycotoxin prevention and control” was published [77] and is a reference for the HACPP plan drafting, but there is a lack of HACCP plans specific for the feed chain in this text. The most crucial points of HACCP for mycotoxin control in the feed chain are reported in Figure 3.

In order to deal with the preliminary stages in commodity flow, an effective team, made of microbiologist; mycologist; and experts in farming, storage, distribution, and trading should be formed. In a holistic approach to mycotoxin control, there is a need for a commodity flow diagram (CFD), which includes all aspects of primary production, drying, storage, transport, and final processing steps (Figure 1). The modern commodity supply chain complicates the creation of this document due to the fact that products move between several owner groups (farmers, traders, transporter, and processors). The commodity and the final product type, climatic zone, and production country will affect the drafting of this report.

Control parameters for the manufacturing of mycotoxin susceptible commodities involve harvesting time, temperature, storage and transportation moisture level, selection prior to processing of agricultural feedstuffs, decontamination environments, and final product storage and transportation [81].

An HACCP plan to manage mycotoxin hazard in feed chain would guide experts at every stage of the supply chain, for which a guide model for the feed chain is reported in Table 2. It is paramount to define critical the control points, hazards, control limits, preventive actions, monitoring strategies (measurable parameters, methods, control frequency, and responsible figure), corrective actions, records, and finally the verifications for each feed chain step. These parameters are certainly specific for each operation and depend on the kind of risk involved.

## 5. Strategies for Mycotoxin Control

There are several well-known strategies commonly used for mycotoxin prevention/decontamination. The main suggested strategies for decontamination include elimination of mycotoxins from grains, decreasing the mycotoxins bioavailability in the animal gastrointestinal tracts, or directly degrading mycotoxins in feeds. Mycotoxins in the feed industry have been extensively studied, especially the prevention and decontamination techniques, on which there are many reviews in the literature [82,83]. The main prevention/decontamination strategies most commonly used in the feed supply chain are reported in Table 3. An update on prevention/decontamination strategies for each feed chain phase, integrated into the design of an HACCP plan, has not yet been carried out.

## 6. Novel Strategies for Mycotoxin Control

The increased request for innovative methods for controlling mycotoxin infections, intoxications, and diseases without keeping toxic chemical residues in the food and feed chain has led to some modernizations. Mycotoxins are studied far fewer from a nanotechnological point of view, although this sector is boomingly evolving; the essential features are based on the nanoscale, which altered the already discovered properties and the realization of infinitely new materials with sensational possibilities. Nanotechnologies and other innovations are closely linked to global trade and more sustainable agriculture that society is demanding nowadays. Modern strategies that manage mycotoxins must cope with climate change, trying to reduce greenhouse gas emissions and resource preservation. Agriculture and livestock must meet the challenges of sustainable development through waste reduction, decreasing energy consumption; encouraging the use of renewable energy, such as solar panels; and moderating water consumption and chemical inputs used in favour of biological substances or methods [111].

To this end, some innovative approaches from mycotoxin prevention to decontamination are reported below, also highlighting at which feed chain level they should be applied to maximize their effectiveness. Given the feed chain integration, some innovative strategies may be appropriate for more than one feed chain step. Innovation can impact mycotoxin management systems and the adopted HACCP plan, providing input for verification, analytical methods, parameters to be used as a reference, and critical control points.

### 6.1. Crop Phase: Pre-Harvest and Harvesting

Modern agricultural technologies, such as precision farming techniques, allow for crop condition control and avoid nutritional and environmental deficits [112]. The rapid development of information technologies has carried to mathematical models (climate models) for regional mycotoxin prediction on the field scale. Climate models, sensitive mycotoxin risk assessment models, and risk maps for predicting mycotoxin contamination with respect to the developed models for predicting crop diseases are still in their primary stages due to the moderately low accuracy [113]. There are few studies on these topics, but it may become necessary to address them in the future as well as to place a focus on climate change and how it will impact mycotoxins [69,114,115,116]. Farmers could use predictions for decision making, though the most accurate predictions available are at the critical growing stage. The feed and food industry can use these technologies for their inspections and decisions (batches buying, routing, and processing). Food safety authorities may improve inspection efficiency based on mycotoxin predictions in constructing their monitoring plan. An interdisciplinary approach is fundamental to make predictive models a support to reduce mycotoxin contamination of food and feed [64].

Biotechnologies are among innovations for mycotoxins control in the pre-harvest phase. To prevent field mycotoxin contamination, a combination of strategies such as bio-competitive fungi and enhancement of host-plant resistance is required [117]. New legislative efforts to limit the use of chemical pesticides have been an encouragement for developing new bio-pesticides and bio-stimulants [118]. Biological control agent (BCA) modes of action are antibiosis, competition, mycoparasitism, and stimulation of plant defence. The studies conducted have shown that the combination of multiple antagonists led to effective results in terms of mycotoxin reduction. Plenty of BCAs have been tested in vitro, but not all of them were effective in the field [119]. This difference in BCA performance could be related to factors such as meteorological metrics, soil properties, nutrient availability, and microbial community. Other important parameters cover the delivery way to the host, delivery form, application time, and application route to provide the interactivity of BCAs against the pathogen. Clarifications are still needed from the legislative point of view.

Another innovation field under study is the antibody-mediated technology. It consists of the design of monoclonal and recombinant fungal-specific antibodies expressed by plants, which can reduce the fungal pathogens spread in the field and can decrease the mycotoxin-production load [120]. Transgenic antibodies have also been studied with positive results. Nevertheless, their maintenance and production are difficult due to the high cost, specific cell culture, and facilities required.

The innovations of pre-harvest phase concerns plant genetic engineering as a biotechnology advance in controlling mycotoxin. Plant genetic engineering for mycotoxin resistance as well as their detoxification place an emphasis on the most economically relevant fungi and mycotoxins [121]. Biochemical and genetic resistance markers have been recognized in crops, being selectable markers in breeding for resistance to mycotoxin infection. Prototypes of genetically engineered crops that enclose two different types of genes have been developed. One for resistance to the phytotoxic effects, thereby helping reduce fungal virulence, and the other contains genes encoding fungal growth inhibitors for decreasing fungal infection, otherwise reducing mycotoxin production by interfering with the biosynthetic pathway [122]. Whereas no one gene has yet been found that can confer crop resistance to infection by mycotoxigenic fungi or to detoxify mycotoxins in plants, there are genes that have the potential to reduce fungal presence and toxin production [117]. The resistance properties in genetically modified plants have been tested under controlled laboratory or greenhouse environments. However, these have not been proven widely in the field [123,124]. Genomics and proteomics technologies can be applied to verify the mycotoxigenic fungi growth and their mycotoxins production in food and feed transgenic crops as well as marker-assisted breeding technologies. To identify genetic control elements, the design of plastid transformation procedures for oilseed/cereal crops also empowers high-level multiple resistance gene expression in the transgenic crop as well as decreases transgene outcrossing.

Nanotechnologies have the potential to revolutionise some conventional methods for mycotoxin control, and they are also spreading in the diagnostic and detection fields. A variety of nanomaterials have been engaged for antifungal or inhibition of mycotoxin [125,126,127]. Nano-antifungal agents are divided in two classes: antifungal mixtures encapsulated into polymeric nanomaterials delivered under proper conditions (e.g., pH variation, higher temperature, and presence of enzymes) or nanomaterials that present fungi inhibition functions. The former has high instability in air, and the latter principally is based on stability, the possibility in green preparation, and the high performance of metal nanoparticles (NPs) [128]. Metal NPs have been assumed to be the most hopeful antifungal agents, such as nano-sized silver ions, which have antimicrobial properties, antifungal activity, and potentiality to control spore-producing of fungal plant pathogens and present economical manufacturing [129]. Nano-fertilizers and nano-fungicides have been experimented in recent years [130]. Foliar applications of engineered NPs are valuable for their potential to enhance plant resistance against pathogens, even if other distribution methods are studied [131,132,133,134,135].

Finally, some nano-diagnostic kits are in phase of investigation; they can be used in the field to detect plant pathogens or to prevent disease development in crops. These myco-sensor assays enable the real-time detection of most common mycotoxins present on a single strip of cereals at or below their EU maximum residue limits. These kits have the advantage of being rapid, cheaper, and easier to use [135,136,137,138].

Enriching and expanding the genetic repertoire of plant secondary metabolites could help increase a plant’s defence systems [117], and biological control of mycotoxin fungi, pests control, investigation on new tolerant hybrids/varieties, targeted and dosed use of chemical products [118], disease forecasting models, and decision support systems can lead to a successful system in order to reduce the mycotoxins problem in feed crops in a sustainable perspective. As seen, in the pre-harvest phase, it is difficult to apply HACCP due to a lack of standardization, climate variability, and different cultivated species. Innovations, however, can also provide solutions for a precise and targeted planning of cultivation operations as well as on the timeliness of harvesting, irrigation, and nutritional interventions for avoiding plant stress and enabling easy integration into the HACCP plan.

### 6.2. Storage

Mycotoxin contamination can occur in farms after harvesting or during the storage process. Improper storage of grain leads to higher quality and quantity losses. Storage grain losses account for 10% to 20%, as a result of insect damages, and nearly 420 Mt of grains are wasted during storage yearly [139]. A valid strategy for mycotoxin prevention in this critical phase is facility-controlled atmospheres (silos and warehouses); in particular, the high N_2_ concentration has highlighted the in situ scientific evidence of a large-scale, eco-friendly, and low cost prevention method for stored cereals [140].

Grain contamination is due to insect and microorganism occurrence and growth, which depend on environmental influences. If a grain is infested, volatile compounds accumulate and emit a smell. Sensing technology allows for the early and accurate detection of insects/fungi. Innovations in this field include technologies used to monitor the quality of stored grain, which provide an automated, rapid, cost-effective, and accurate method. To observe the quality and to detect the off-flavour related to contamination, some technologies are available including environment, acoustic, and odour sensing through e-noses and image sensing, which could improve the storage facilities [141].

A valid storage system using information technologies could ensure grain quality by controlling and monitoring environmental factors such as temperature, light, humidity, pests, and hygiene in order to reduce feed wastage. Gas, electrostatic, and acoustic sensors for particle size detection of grain dust/moisture are suitable as instruments for improving stored grain management. Wireless sensors integrated or installed separately inside grain silos and communicating with each other in a network approach could provide useful data for monitoring grain storage conditions [142].

Real-time monitoring systems improve the quality of storage and reduce not only the losses but also work force and labour intensity. Precision monitoring systems of insect population and application of control doses of insecticides ensure grain safety and security [143,144,145,146]. The next generation storage systems not only control equipment and collect data but also are integrated into the company’s animal feed safety and quality systems or HACCP plan, allowing for control and troubleshooting from a smart phone.

### 6.3. Feed Mill Operations

A coordinated mycotoxin feed mill management program starts with wide-ranging quality assurance. It must include more than merely testing for mycotoxins during the receiving process, and it must have strategies to preclude their introduction into the manufacturing process. To ensure the production of high-quality feed, a team effort that requires the adherence of top management, feed mill manage, and employees is needed; this can lead to a safe, clean, and well-maintained mill. The primary element in the incoming screening program is to evaluate the risks connected with each ingredient, so the end feed product is within the limits fixed for various species. The feed production operations are complex and include many passages, such as screening, sorting, analyses, and various technological treatments before packaging. Feed manufacturing operations may vary, relying on the species for which it is intended and the type of feed produced (mash, crumbled, pellet, complete, or supplements).

#### 6.3.1. Innovative Detection and Sorting Techniques

Accurate sampling and testing is crucial for a comprehensive mycotoxin feed management plan. In view of an HACCP plan, a targeted sampling plan should be provided for each product, specifying the technique, the instrument, the operator, and the frequency with which they are to be made and recording all of the results. The sampling must be representative for the raw materials analysed; therefore, evaluating and planning appropriate purpose-based sampling plans are of extreme importance [147,148]. With regard to feed, the European legislation has published a consolidated text [10], which establishes the methods of sampling and analysis for the official control of the mycotoxin levels in foodstuffs.

The need for fast test results is even more essential. Sampling and testing requires personnel and lab equipment and needs to fit into an existing schedule. A number of methods for fast analysis, including rapid test strips, is increasingly complementing chromatography-based methods. The most popularly used in the food industry are immunoassay-based methods and biosensors [3,149] although publications targeted at feed industry are scarce. In detail, the former could be dipsticks, flow through a membrane-based immunoassay, a immunochromatographic assay, a fluorometric assay with immunoaffinity clean-up column or with a solid phase extraction clean-up column, and a fluorescence polarization method. Although issues in test strip elaboration (insufficient sensitivity, selectivity, or strong matrix dependence) can remain, high-quality ones include completing conventional detection methods. The major requirement for easy-to-use assays is to obtain good antibodies because these simplest technologies present lower sensibility.

The development of innovative test strips leads to legislative specifications for maximum acceptable levels of food/feed mycotoxins through fast screening [150,151]. Some emerging technologies in mycotoxin analysis and detection are reported in Table 4. The innovations listed in the table, in some cases, referred to studies carried out in the food chain, but given the integration of the feed/food supply chains, these techniques can be considered a model for the feed industry. These technologies are available on the market, but for the feed industry, their inclusion must be assessed according to the specific requirements of the sector.

Future challenges in mycotoxin testing are multi-mycotoxin assessment, further miniaturization, and portability for on-site mycotoxin testing. In addition, these methods should be user-friendly and cost-effective.

Microfluidic “lab-on-a-chip” devices might incorporate and miniaturize functions from sample preparation to detection, revealing propensity in rapid, accurate, and high-throughput detection of mycotoxins. These devices have a great potential for accurate and high-throughput detection of mycotoxins in agro- and food products [189]. These innovative methods could transform complicated conventional methods into simpler micro-scale devices. The feature of portability enables the development of in-field devices. The final objective for commercial detection of mycotoxins in agricultural and food products is the integration into reusable microfluidics systems. Efforts are also necessary to develop user-friendly devices, reducing potential risk to operators and the environment [188,190].

Current detection methods for mycotoxin contamination utilized in the cereal industry rely on bulk level. However, mycotoxin contamination is unbalanced and bulk samples do not always constitute the whole batch contamination. Single kernel assessment can offer insight into mycotoxin contamination as well as can propose alternatives to the inaccuracy in bulk detection. Liquid chromatography, fluorescence imaging, and reflectance imaging may detect mycotoxin at a single kernel level; this might provide improved remediation methods through sorting, which could affect feed security and feed waste administration [191].

#### 6.3.2. Innovative Decontamination Techniques

The modern direction is towards non-invasive decontamination methods, which do not alter the chemical composition and the quality of the raw materials but which have a certain effectiveness in reducing the presence of mycotoxins. Many of these techniques are diffused in the food industry, but studies in the feed industry are few or non-existent. Therefore, their inclusion in the feed mill or in the feed production chain have not yet been evaluated. The feedstuffs mycotoxin decontamination takes place by blocking their production by fungi or through their detoxification from matrixes. With regard to detoxification methods, they can act by removal through mechanical processing steps or degradation by chemical agents, physical methods, or biological treatments [192].

Ozone application has shown auspicious outcomes for mycotoxin management in the food industry and is becoming of interest in the feed chain. Ozone is an easy technology, and its application in food is considered safe and effective enough to be recognized as green technology by United States Environmental Protection Agency (EPA) [193]. In particular, O₃ inhibits fungal growth, sporulation, and germination, offering insignificant nutrient loss or sensory properties in food/feed. However, its antimicrobial activity depends on the vegetable/fungus species, growth stage, concentration, and exposure time. The obstacle at the moment seems to be the determination of the degradation products that form after O₃ treatment. In vivo and in vitro toxicological tests should be performed to evaluate the consequences of degradation products on human and animal health [194]. In addition, for powerful and safe processing, suitable O₃ concentration, contact time, and other conditions should be distinct for food/feed [195].

Cold plasma technology reduces and degrades mycotoxins in/on food and feed materials, and it might be a sustainable method that requires lower energy input and investment. Several aspects of cold plasma technology in reducing pathogenic fungi and mycotoxins in/on food and feed materials need to be defined. Cold plasma treatments must handle the bulk quantities of food and feed, but more tests are required. Cold plasma treatment efficacy depends on several factors: surface characteristics, type of food and feed, mycotoxin nature/structure, fungi type and their surface attachment, fungal ability to spread on surfaces, time of treatment and life time during treatments, cost-efficiency, etc. Cereals and feed mycotoxin can be eliminated using plasma treatments without inducing relevant modification in the nutritional composition. The consequence of cold plasma on mycotoxins in flour and the related quality modifications have not been evaluated [196]. In the field of cereals, some studies have been carried out, which report AF reduction from 62% to 93% according to the type of cold plasma used and contact times [197,198,199,200,201,202].

A recent innovation in the area of mycotoxin decontamination is the use of electromagnetic radiation, which includes gamma radiation, pulsed-light, radio frequency, and microwaves. Gamma radiation has been evaluated as an effective method for maintaining agro- and food quality [203,204]. They destroy pathogenic microorganisms, causing direct or indirect (production of free radicals and ions) damage to DNA in microbial cells [205,206]. The results of gamma irradiation for mycotoxin reduction are conflicting, as its efficacy depends on the number and type of fungal species, food composition, radiation dose, and air humidity [206]. Pulsed light has also been applicate to mycotoxin decontamination, generating short, high-intensity flashes of broad-spectrum white light. The combination between full spectra of ultraviolet, visible, and infrared light eliminate the cell wall and nucleic acid of mycotoxin fungi present on the surface of feed/food or packaging materials within seconds [207]. Radio frequency and microwave are valid alternative methods for reducing mycotoxin contamination in agricultural products. The effect of microwave on AF-contaminated maize resulted in 36% reduction of AFB1 and 58% for AF-B2 in a 5.5 min treatment [208].

In recent years, nanomaterials have found a leading role in the photocatalytic degradation of mycotoxins. They are low cost, environmentally friendly, and easily applied with no secondary pollution [209,210,211,212]. Nanomaterials such as graphene/ZnO hybrids, Fe_2_O_3_, and titanium dioxide (TiO_2_) have been used for mycotoxin photocatalytic degradation [212,213,214].

#### 6.3.3. Packaging Innovations

The fundamental function of an optimal packaging material is to ensure the quality of food/feed during transportation and storage, thereby extending the product’s shelf life. Active or improved packaging consist of an interaction between the package and the product, which limit the growth of microorganisms and reduce quality deterioration processes [215]. These innovative technologies have already been studied in the food sector, but currently, publications in the feed industry are scarce.

The existing types of active packaging that could control mycotoxin formation are moisture regulators and antimicrobial active packaging. The former uses moisture absorbers; the latter are based on carbon dioxide, ethanol, preservatives, inorganic nanoparticles, and natural extracts. Antioxidant and oxygen scavenger packaging can also reduce microbial growth, maintaining oxidative stability, as well as thermal insulation packaging based on phase change materials [216,217]. In this direction, given the growing need for plastic reduction, bio-based packaging is being developed also to reduce foodborne pathogens [218].

Nanofibers have acquired significant interest in the packaging industry. Nanotechnologies existing in packaging include biodegradable nano-sensors for temperature and moisture monitoring, nanoclays and nanofilms to prevent spoilage and oxygen adsorption, and nanoparticles for antimicrobial and antifungal topcoats [219].

Two hybrid nanofibers were developed using both synthetic and natural polymers as antifungal food packaging materials [220]. They consist of cellulose acetate or polyvinyl chloride (PVC) encapsulated with silver nanoparticles (AgNPs) set up by electrospinning. The AgNP incorporation in both materials prevent yeast and mould growth, resulting in promising feed antifungal packaging. In this direction, nano-packaging is in continuous development, as demonstrated by the number of nanocomposites patented from 2012 to 2017 [221,222]. Furthermore, the Food and Drug Administration (FDA) has endorsed ZnO and TiO_2_ in food packaging [223], but the use of carbon nanotubes has not been approved yet. Other information about polymer-based nanoparticles packaging and methods of incorporation have recently been published [224,225].

Another innovation, with action on the products, is intelligent packaging able to monitor the condition of the products and to provide information. Available systems use indicators (time–temperature, freshness, and gas indicators), sensors (chemical, biosensors, or edible sensors), or data carriers (barcode labels and RFID tags). Next-generation packaging will track ingredients from farm-to-fork and from the time of purchase, during receiving, processing, and delivery to the animal producer by the Integrated Radio-Frequency Identification (RFID), already widely used in the food supply chain for traceability and for livestock [226]. This could be a valuable aid tool for mycotoxin management in the feed chain, and it would be added to ingredients and finished feed trucks [188,227]. The union of active and intelligent packaging allows for the development of smart packaging, which may have importance in the near future, and it can simplify the monitoring and management of feed in an HACCP perspective.

Finally, Purdue Improved Crop Storage (PICS) bags are an important packaging innovation especially in developing countries, where technologies are not so advanced and available. They consist of a triple bagging hermetic technology with two inner liners of high-density polyethylene (HDPE) and an outer layer woven PP. PICS bags can decrease the oxygen influx and can limit the emission of carbon dioxide, which prevents insect growth in stored grains. Several studies have reported that PICS bags prevent fungal growth [228].

### 6.4. Livestock Production

Although feed manufactures ensure that low-mycotoxin feed leaves the mill, time is expended in storage on the farm preceding animal consumption. The feed miller should provide rigorous information related to feed storage, including mould inhibitors, and should use the best packaging available, although the management in this case is up to the farm.

Mycotoxin detoxification is deemed constant challenge in the feed industry [229]. The most effective approach to minimizing the adverse impact of mycotoxins in livestock is the use of mycotoxin adsorbents. There is a huge amount of literature regarding this topic (Table 3—livestock production). Inorganic binding agents can be improved because they present relative specificity and might bind essential nutrients, although they may accumulate in the environment, leading to soil and pasture pollution, and may show relative effectiveness except for AFs.

The organic binders and the bio-transforming agents are degradable and eco-sustainable, and they have low inclusion rates, multiple-micotoxin binding ability, economic feasibility, high specificity (enzymes specific for each mycotoxin), and no toxicity. These trends act as a vector for innovations in this sector, which ensures high animal performance, maintains their good health, and guarantees the quality of derived products, respecting the environmental sustainability principles and avoiding substantial economic losses due to the use of drugs, feed, and derived animal products waste [230].

Biotransformation may be the prospective of mycotoxin risk management; it transforms non-absorbable mycotoxins into innocuous elements without risks for livestock. In addition, bio-protection is gaining interest due to scientifically proven plant and algae extracts blend that stimulates animal liver and immune function in order to address the adverse consequences of mycotoxins.

Following these principles, the innovative actions in the last decade are reported in Figure 4.

Combined mycotoxin binders such as bentonite, which includes micro-organisms, enzymes, and a blend of plant and algae extracts for hepato- and immune-protection have been tested. The inorganic and organic adsorbents mixture may be suited to the multi-contaminated feeds, and these forms could combine adsorption, biotransformation, and bio-protection, offering the most complete covering against the adverse impact of mycotoxins [231,232].

Micronized fibres can be useful as mycotoxin adsorbents thanks to positive gut adsorption and greater faecal excretion [233]. They are cellulose, hemicellulose, and lignin, derived from different plant such as cereals or legumes (wheat, alfalfa, oat, barley, pea, and hulls). At first, these fibres were focused on micronized wheat fibres that showed positive effects against OTA adsorption [233], but recently, this field has evolved in a new direction: agro-by-products. Red wine waste including dehydrated grape pomace (rich in phenolic compounds) has been revealed in vitro to remove mycotoxins in a liquid medium (AFB1, ZEN, OTA, and FBs), being a promising excellent adsorbent [104,234]. Apple pomace (rich in fibres and pectin) was proven as a mycotoxin adsorbent in pigs incorporating it in DON-contaminated feed, and the adverse effect of DON may be reduced [235]. Another study uses grape steams and olive pomace that demonstrated in vitro efficacy [236]. Bio-sorption through agricultural by-products has been established as a low cost and safe way to reduce mycotoxins. Agricultural by-products can simultaneously adsorb a wide range of mycotoxins; a recent study, compared 51 agricultural by-products for their capability to adsorb mycotoxins, and the results showed that grape pomaces, artichoke wastes, and almond hulls were promising bio-sorbents for mycotoxins, effective against AFB1, ZEA, and OTA. These studies showed promising technological applications for agricultural by-products as feed/food additives for mycotoxin reduction applying the principles of circular economy and reducing the food chains waste through a valorisation of these products [237]. Discovering new technologies to utilize antifungal compounds from food/feed wastes or co/by-products, such as olive oil wastewater or winery by-products, may enhance sustainability and may reduce costs.

Humic acids originate from the natural decay of organic plant materials, and their in vitro studies have shown interesting insights. They can adsorb mycotoxins, especially AFB1, OTA, and ZEN [238,239]. In a recent study, humic acids (0.1 to 0.3%) were used as an aflatoxin binder in broiler chickens, and the results showed that they could alleviate Aflatoxin toxic effects in growing broilers and might be positive for Aflatoxin-contaminated feedstuff poultry management in association with other mycotoxin management practices [240], extending the concept to other animal species.

Some chlorophyll-based adsorbents as potential sequestrating agents against mycotoxins have been studied. They are Chlorophyllin or other substances that act by trapping the planar structure of the mycotoxin ring, inhibiting the link between the toxin and DNA and thus facilitating their excretion [241]. From the preliminary studies, they seems to be valid in combination with other adsorbents, inorganic, and valid for ruminant species administration.

Among the organic adsorbents, there is activated carbon, which is an insoluble powder, originating by pyrolysis of some organic compounds, succeeded by its physical or chemical activation in order to develop a high porous structure. It has been tested in vitro, demonstrating a certain effectiveness that in vivo has yet to be confirmed. They present some advantages such as economic viability, plant and organic source origin, valid commercial alternative in the market, and eco-sustainability [242].

Adsorbents of synthetic origin are however widespread, with evolutions compared with conventional ones. Organo-aluminosilicates or modified clays, through modifications, can exceed their limit of selectivity towards a single mycotoxin. Modifications consist in altering their surface characteristics by switching the structural charge-balance cations with high-molecular-weight quaternary amines, increasing their hydrophobicity. Several studies have reported a greater efficiency of organically modified clays compared with conventional ones [243]. Other adsorbents agents of synthetic origin are polymers. Cholestyramine, divinylbenzene-styrene, and polyvinylpyrrolidone are some examples that have been shown to bond mycotoxins in both in vitro and in vivo experiments. In particular, the former is an insoluble quaternary ammonium exchange resin that bonds various anionic substances and weakly adsorbs neutral/cationic ones by nonspecific binding. Their limitation could be the higher costs for practical applications, although they present a higher efficacy [84,244].

Growing interest is targeted at biological detoxification approaches based on competitive exclusion by non-toxigenic fungal strains for mitigating mycotoxin development and precluding their absorption into the animal body. This is increasingly defined as microbial detoxification. Various microorganisms, such as lactic acid bacteria (*Lactobacillus*, *Bifidobacterium*, *Propionibacterium*, and *Lactococcus*) are active as mycotoxin binders [245,246,247]. The binding is a surface phenomenon with the contribution of lactic acid and other metabolites such as phenolic compounds, hydroxyl fatty acids, hydrogen peroxide, reuterin, and proteinaceous composites produced by lactic acid bacteria [248]. Other microorganisms were carried over to bind/degrade AFs in feed/food such as *Saccharomyces cerevisiae* and some bacteria such as *Rhodococcus erythropolis*, *Bacillus* sp., *Stenotrophomonas maltophilia*, *Mycobacterium fluoranthenivorans*, and *Nocardia corynebacterioides.* The results reached in mycotoxin microbiological decontamination may be considered a stepping stone in the design of commercial technologies. Further microorganisms screening can result in the finding of effective microbes. With the application of molecular biology techniques, mycotoxin degrading microbial species may be developed to increase feed quality and safety from mycotoxins protecting animals’ health. Plenty of interrogatives need future tests for optimal timing, pH, inactivating bacterial methods, and lactic acid bacteria concentrations that would achieve greater findings. Future trials should also concentrate on detecting the mycotoxin-binding mechanisms to ensure that it is permanent. With the right directions, lactic acid bacteria may be applied widely in the manufacturing of raw feed/food liable to mycotoxin contamination [249].

The enforcement of plant compounds to prevent toxigenic fungi and mycotoxin production has been examined frequently in vitro in the last decade. The action mechanisms target the cell wall, the plasmatic membrane, proteins, and the mitochondrial functionality of fungal cells. Some elements also function as mycotoxin biosynthetic pathway down-regulators, whereas others have a direct degrading activity through mycotoxin [250]. The in vitro fungitoxic properties of essential oils rich in phenols; terpenes and N-containing compounds; and secondary metabolites such as tannins, terpenoids, alkaloids, and flavonoids have been demonstrated. In vivo trials often exhibit low reliability due to the not fully characterization of essential oils composition and active compounds quantification; the impact is not completely determined because there can be variation in gastrointestinal tract anatomy and functionality within the same species [251]. Plant extract antifungal action has great potential thanks to their easy preparation and safety. Furthermore, they are efficient in view of their systemic action and lack of residual impact, biodegradability, and enhancement of animal metabolism [252,253,254,255]. Among the most recent innovations for bioactive plant components/metabolites are their synergy with nano-mycotoxin absorbing agents in order to reduce mycotoxins and their adverse health effects. Β-cyclodextri-based nanosponges encapsulated with plant bioactive compounds, designed to combat toxigenic fungi, reduce mycotoxins in feed/food, without health and environmental risks for animals and humans [256].

The evolution demanded from the market and the reduction in resource consumption have allowed for nanotechnologies to spread against mycotoxins in livestock production as well [160] (Table 5).

Table 5 shows a first attempt at classifying nanotechnologies used for the prevention of livestock contamination against mycotoxins and includes both organic and inorganic compounds. Nanomaterials have the characteristics that market is looking for: high specificity, high porosity, green production and origin, and economy. Hybrid nanomaterials may be used as mycotoxin-detoxifying agents and act as mycotoxin binders. Carbon nanomaterials, chitosan polymeric NPs, nanoclay binders, antibodies or targeting peptides, and magnetic Fe_3_O_4_ modifiers have been largely utilized for mycotoxin adsorption [270,271].

Several hybrid nanomaterials can sequester mycotoxins, such as silica-based inorganic substances (i.e., natural clays or synthetic polymers such as calcium aluminosilicates, bentonites, hydrated sodium, zeolites, etc.), carbon nanoforms, and other polymers (cholestyramine, divinylbenzene-styrene polymers, polyvinylpyrrolidone, and humic acid polymers) [128]. Carbon nanoforms (graphene, graphene oxide, fullerenes, fibres, and nanotubes) are amphoteric and their protonable and de-protonable surfaces render them optimal as mycotoxin adsorbents. Carbon nanomaterials (e.g., nanodiamonds, CNTs, and magnetic graphene (MGO)) have been applied for mycotoxin adsorption for their adsorptive capability, large surface area per weight, high stability, inherent inertness, and colloidal stability in diverse pHs (Horky et al., 2018). The high nanomaterial expectations to degrade mycotoxins relies not merely on the high surface area and the organic compounds affinity but more on their potential to be designed to increase the target mycotoxin selectivity [128,129,271,272]. A series of in vitro studies show that the potential is achievable; however, the lack of in vivo studies in the case of animal applications is another aspect to be taken into account.

Mycotoxins can lead to increased oxidative stress in organisms. A viable strategy to reduce these effects is nutritional supplementation. Superior doses of natural antioxidants supplements just as tocopherol, selenium, or zinc are advised in animal feed as well as vitamins A, E, and C; folate; provitamins; carotenoids; and selenium, and the combination of multiple substances can promote the fight against mycotoxins. The combined use of antioxidants, yeasts or selenium-enriched yeast, calcium and magnesium, limiting essential amino acids (Methionine, Alanine, or precursor amino-acids), and fats can be beneficial during mycotoxicosis [230].

Cerium, zinc (titanium) oxide, silver, gold, selenium, or carbon-based nanoparticles have proven antioxidant features. The improved antioxidant ability of metal nanoparticles might be achieved by green synthesis; they are enriched with natural compounds, which increases their donor activity. Polymeric nano-capsules preserve and release antioxidants to the target tissue.

Mycotoxins are non-proteinaceous and have low molecular weight. Their application as conjugate immunogens should be explored due to the toxic characteristics of the molecule that might be delivered. A possible vaccine might be the protein conjugates of “mycotoxoids”, chemically detoxified mycotoxins that can induce antibody responses. Vaccination may be a viable opportunity for preventing animal risks caused by mycotoxins. This vaccine production is costly and not in use as a mitigation strategy [273,274].

The major challenges that innovations in livestock production have to exceed are the active compound characterization; the dose standardization; the biological activity; the interactions with the feed matrix, feed ingredients, or other feed additives; and the toxicological characterization of the degradation products. Moving from research to in-field applications, we have to consider the legislative requirements. The application of new mycotoxin-reducing agents as additives needs the approval of governmental authorities, and they undergo a scientific assessment by EFSA [275], although some have already been tried and are commercially available.

## 7. Conclusions

The feed chain is very complex, and continuous research is necessary to improve mycotoxin management. A single approach for this contamination is not enough; an integral chain system that includes all the steps involved is required. The HACCP cannot be expected to represent a panacea for mycotoxin control, as the risk of contamination is unavoidable. Global adoption of GAP, GMP, and HACCP standards developed to mitigate mycotoxin exposure are the basis for designing an effective HACCP-based integrated mycotoxin management program that must take into account factors such as climate, farming systems, pre- and post-harvest technologies, public health, producer and processor conformity, accessibility of analytical tools, and the economy [276]. In order to be effective, these management systems have to communicate between experts in the whole value chain, comprised of farmers, consumers, processors, and traders. With this procedure, each production phase could potentially decrease the risks, and the final product minimized hazards associated with mycotoxin contamination.

As mycotoxin co-occurrence, and modified and emerging mycotoxins are the heart of modern mycotoxin research, some advanced management strategies that can potentially decrease this contamination in feed chain have been reported by this review, also taking into account the effects of climate change.

Biological and nanotechnological innovations are promising alternatives to conventional methods for a sustainable development in which lower use of inputs and resources is required while maintaining high production and yields. Similarly omics techniques, molecular techniques, sensing technology, and other innovations such as information systems and mathematical prediction models provide a strong boost to the feed industry and related sectors such as the livestock, agriculture, and food sectors. Future research should be focused on methods to analyse multi-contaminated samples and to find specific analyses for emerging mycotoxins, including modified forms. Recent insights have generated new relationship between mycotoxins and ruminal/gut microbiota, presenting that gut microbiota could be implicated in mycotoxicosis development. These insights can drive the design of improved approaches for the prevention and therapy of mycotoxin contamination in livestock production [277,278].

Controlling mycotoxin contamination through the enforcement of global gold standard could prevent health risks [279] in a number of ways: limiting mycotoxin residues in animal origin food to ensure public health, reducing waste and losses of raw materials (agro-commodities), improving the efficiency and quality of crops produced and the resulting feed, simplifying the production of safe feed, ensuring and improving animal health in an economic and environmental sustainable way. Awareness of these advantages can encourage policy makers and value chain actors to consider operative ways to manage mycotoxin in the whole feed chain.

## Figures and Tables

**Figure 1 toxins-13-00572-f001:**
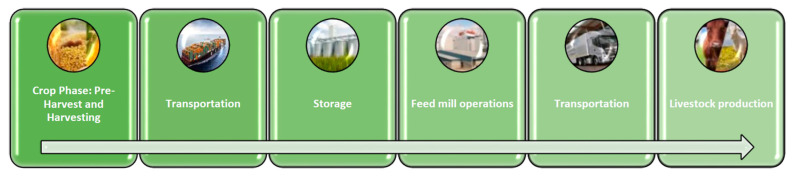
Representation of the feed chain.

**Figure 2 toxins-13-00572-f002:**
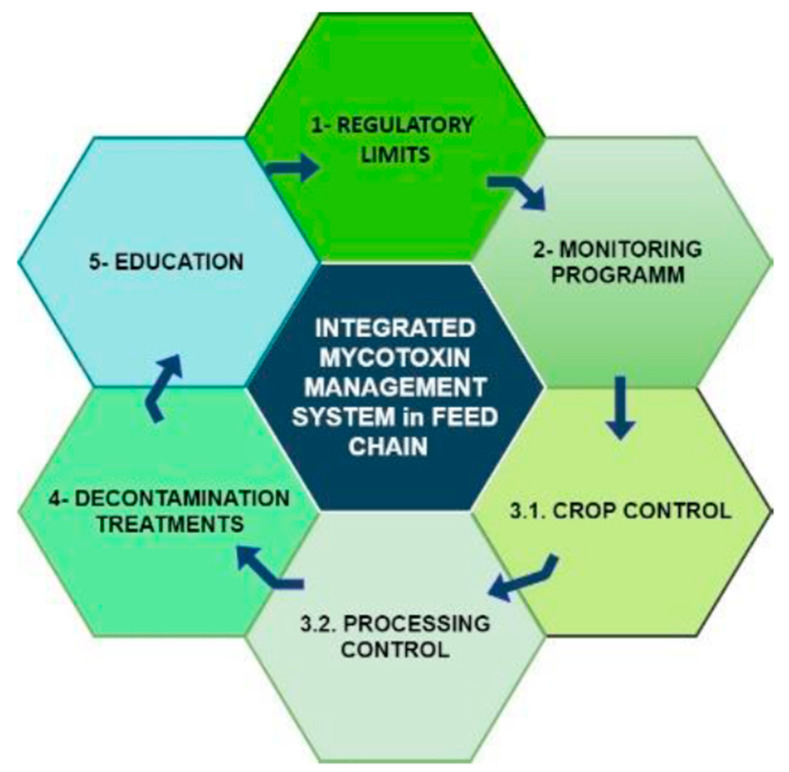
Integrated system phases for mycotoxin management (modified from FAO 1995).

**Figure 3 toxins-13-00572-f003:**
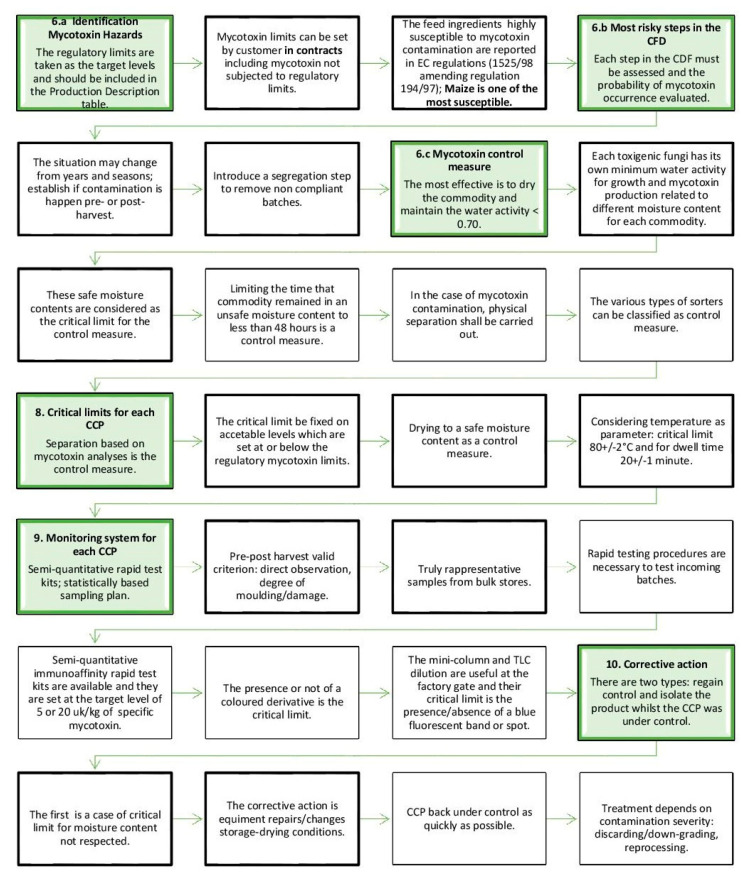
Crucial HACCP tasks for mycotoxin control in the feed chain (modified from FAO, 2001).

**Figure 4 toxins-13-00572-f004:**
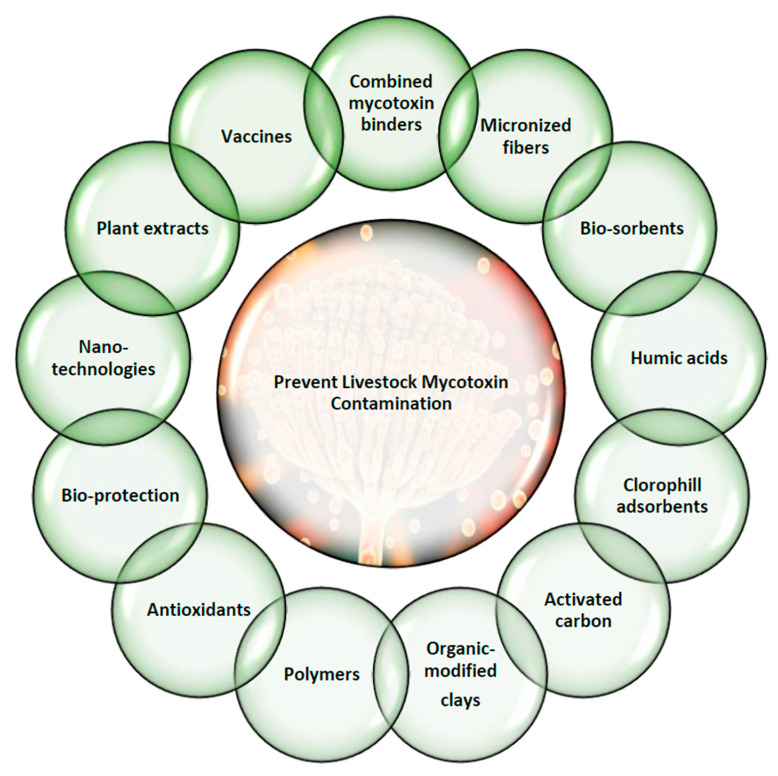
Novel strategies for preventing livestock mycotoxin contamination.

**Table 1 toxins-13-00572-t001:** Phases of an integrated mycotoxin management system (modified from FAO, 1995).

Phases of Integrated Mycotoxin Management System	
1. Setting of regulatory limits	- Commodity surveys to identify contamination levels; - Dietary intake surveys to regulate consumption levels; - Toxicological data Assessment; - Establishment of analytical technical knowledge; - Feed stock availability based on specific regulatory limits.
2. Creation of a monitoring programme	- Institution of a sampling plan: (a) sample collection; (b) test quota preparation; (c) test quota analysis; - Permitted procedures of mycotoxin-contaminated products.
3. Crop phase Control 3.1. Processing Control	- GAP; - GMP; - Quality control.
4. Specific decontamination actions	- Final product assessment; - Term of use of treated product.
5. Consumer education/producer training	

**Table 2 toxins-13-00572-t002:** Managing mycotoxins in the feed chain: guide model of a HACCP plan.

CCP	Hazard	Critical Limit	Preventive Action	Monitoring				Corrective Action	Records	Verification
				Parameter	Method	Frequency	Responsable			
Pre-harvest	Low soil moisture/plant stress	Lower limit of critical Aw	Irrigate	Soil moisture value		Weekly on Tuesday morning	Agronomist	Supplementary irrigation	Soil moisture	State of plants
	Insufficient soil nutrients	N,P,K applications	Fertilise	Fertilizer application		As recommended for hybrid	Agronomist	Additional fertilizer	Amounts and type of fertilize	State of plants
	Insect attack	Insect population within accettable limits	Integrated pest management plan	Visual inspection and sample		Weekly	Agronomist	Apply pestice in accordance with IPM plan	Results of the monitoring	State of plants
Harvest	Damage kernel	Moisture content <14%	Harvest when kernels are dry	Measure grain moisture		Prior to harvest	Farmer/ Agro-mechanical	Postpone harvest till kernels properly dried	Grain moisture	Visual inspection/analyses of raw materials
Storage	Excessive moisture content	Moisture content <14%	Do not store until kernels dry	Measure grain moisture		Immediately prior to storage	Commodity quality assistant	Dry mechanically	Grain moisture	Analysis
	Insect attack	Inspection protocols show no evidence of insect or rodent infestation	IPM plan	Visual inspection		Weekly	Mill operators	Follow IPM plan for pest control method	Visual inspection	Analysis results
	High humidity and temperature	Temperature and humidity within limits recommended in industry literature	Aerate grain to control temperature and humidity	Measure humidity, temperature and airflow		Daily during storage	Mill operators	Adjust aeration- time, or airflow to achieve desidered temperature and humidity	Humidity, temeperature and airflow	Authomatic monitoring systems
Feed mill	Increase of myctoxin levels in mixer phase	mixer cleaning mycotoxin levels	Controlling mixer cleaning and way of frequency	ppb	ELISA and UV	Before every mixing process	Feed quality assistant	Changing the time and method of cleaning	Cleaning and disinfection register form	Cleaning, analysis results
	Increased myctoxin levels in Cooler	The heat of feed should be at most 5 °C more than environment heat	Increasing empting time of the cooler; decreasing the capacity of pellet; controlling the heat levels of the cooler	°C	Thermometer	Daily	Foremen	Mixing with cold feed, keeping a backup cooler	Cooler heat follow form	Measuring heat during cooling process
Livestock production	Increase of mycotoxins levels	Temperature, cleaning	Feeding silo cleaning, climate and insect/rodent control	°C, ppb	Thermometer, ELISA	Before every entering livestock, daily, weekly	Farmers, Livestock keeper	Dietary manipulation, on-farm management strategies, use of binding agents	Live activity form	Cleaning and disinfection results, analises of animal products

**Table 3 toxins-13-00572-t003:** Mycotoxin conventional prevention and decontamination strategies applied in the feed chain.

Feed Cycle Phases	Strategies	References
Pre-harvest prevention	❖ Application of GAP and improved soil cultivation practices: proper crop variety, crop rotation (e.g., wheat and legumes), ploughing, minimum tillage or no-tillage, weed control practices; ❖ Crop breeding/use of resistant varieties; ❖ Proper sowing date; ❖ Avoid stressors: adequate irrigation schedules, prevent pests attack enforcing insect control programmes; ❖ Using fungi controlling products (fungicides).	[84,85,86,87,88,89]
Harvesting	❖ Timeliness to reduce moisture or water activity; ❖ Minimum mechanical damage; ❖ Check of crop clean-up and drying; ❖ Remove extraneous materials; ❖ Dry rapidly below 10% moisture and maintain at lower temperature.	[90,91]
Storage	❖ Quality control at intake: grain entering storage must be proper condition of (moisture and disease levels, kernel entirety); ❖ Storage buildings suitability of, cleaning and sanitation limit the accumulation of dust which favours the mould development, make treatments according to ‘Plant Protection Product Manufacturers’ instructions, residues do not exceed authorised levels; ❖ Prevention of infestation into storage facilities by invertebrates and rodents and birds, including insect trapping, using fumigations such as phosphine or essential oils; ❖ Controlling and keep monitoring biotic factors (grain, bacteria, yeast, fungi) and Abiotic factors (water, air, temperature). Recommended internal humidity less than 17–20% (depending on the crops), Water activity inferior to 0.7; High CO_2_ and N_2_ levels and decreased O_2_ levels; Avoid temperature and humidity increase, reduction of condensation inside the storage structure (silo or warehouse); maintenance of storage conditions; ❖ Application of mould inhibitors (acidifiers such as organic acids, antioxidants, or essential oils) in combinations or individual organic acids (sorbic, propionic, acetic, and benzoic acids), salts of organic acids (potassium sorbate and calcium propionate) and copper sulphate; ❖ Grain aeration and grain movement, grain ventilation; ❖ Infrastructure needed to identify and rectify problems rapidly by lowering temperature; infected grains will need to be destined for a new use and not for feed production; ❖ Silos equipped with temperature monitoring and aerations systems can help to temperature control and condensation reduction; ❖ Silo-bags are recommended in the case of temporary hermetic storage when permanent facilities are not available in farms, they are a low-cost alternative. ❖ Correct handling of waste	[92,93,94,95]
Feed mill operations	❖ Grain processing and testing prior to acquisition; Testing raw materials and all the ingredients; ❖ Frequent and regular cleaning of manufacture plants, feed-mill environment, operators, equipment and feed storage facilities; ❖ Processing/manufacturing operation monitored to maintain high quality product: regular manufacturing control procedures and enforcement programmes such as GMP, HACCP and standards; routine inspections; schedule samples and analysis, mycotoxin proficiency and testing program; turnaround time of stored grains; ❖ Physical methods: cleaning and aggressive/intensive sorting (air separators, sieves, gravity separators and indented cylinders, wet flotation, colour-sorting); dehulling; milling; separating the outer seed coat, or bran; ❖ Thermal methods: dry heating, superheated steam, extrusion cooking, non-ionizing and ionizing irradiation; ❖ Biological methods: bacteria, yeast, fungi, other eukaryotic microorganism with detoxification activities and catabolizing detoxifying enzymes and their coding genes, polypeptides such as chitosan; ❖ Avoid broken or cracked grains.	[96,97,98,99,100,101,102]
Transportation	❖ Regular cleaning (vehicles and associated machinery) to prevent cross contamination and treatments with mould inhibitors, especially in “dead spots” areas; ❖ Treatment of raw materials with a mould inhibitor prior to transport in the case of long distances roads;	[63,103]
Livestock production	❖ Using high-quality and selected feedstuffs; ❖ Test products and monitor mycotoxin levels in feed ingredients; ❖ Combining normal to high-quality feedstuff with low cost and quality/high mycotoxin contamination risk; ❖ Feeders on the animal farms must be cleaned periodically with periodic elimination of feed residues; ❖ Dietary manipulation: high nutrient levels, increasing antioxidant levels of the diets, increasing level of dietary selenium, increasing protein level, supplementing with vit C and E; ❖ On-farm management strategies: reducing the exposure time, target feeds to species/animals according to the permitted limits ❖ Preventing mycotoxin uptake by using feed additives: ❖ Binding agents: ❖ Inorganic: sodium calcium aluminosilicates such as phyllosilicates (smectites), tectosilicates (zeolites), activated coat, montmorillonite clays, bentonite; ❖ Organic: polysaccharide-beta glucans and MOS, yeast cell walls and live yeast *Saccharomyces cerevisiae*, plant extracts; ❖ Bio-transforming agents: bacteria, yeast, fungi, and enzymes degrading mycotoxin molecules in non-toxic metabolites.	[63,64,103,104,105,106,107,108,109,110]

**Table 4 toxins-13-00572-t004:** Emerging mycotoxin detection/sorting techniques.

Method	Feed Matrix	Advantages	Disadvantages	Reference
Detection techniques				
LFD—Lateral Flow Devices	Cereal and cereal-based foods, cereal grains	Rapid, no clean-up and expensive kit, user-freindly, no training required;	Semi-quantitative, cross-reactivity with correlated mycotoxins, verification required for extra matrices.	[151,152,153]
FPIA—Flourescent Polarization Immuno-Assays	Cereals	Fast, no clean-up required, confirmed for DON in wheat.	Conflicting with ELISA or HPLC analyses, scarce sensitivity, cross-reactivity, matrix interference.	[154]
Biosensors	Cereals	Rapid, no clean-up procedures.	Cross-reactivity with related mycotoxins, extract clean-up needed to improve sensitivity, variation in reproducibility and repeatability.	[155]
Nano-biosensors	Feed	Rapid mycotoxin detection; Nanomaterials improves bio-sensors properties making them target-specific with enhanced sensitivity and affordable; Applications for mycotoxin assay (HPLC-based assays, nanomaterials-based immunoassays, nanomaterials-based aptasensors, and nanomaterials in MIPs-based mycotoxin sensors).	Further studies needed for existing applications; Development of nanobiosensors to analyse and detect multiple mycotoxins.	[71,156,157,158,159,160]
IR spectroscopy (FT-NIR and PCA; UV-vis spectroscopy; NIR-RS + MSM)	Wheat, maize	Rapid, non-destructive measurement, no extraction or clean-up, easy operation.	Expensive equipment, calibration model need validation, knowledge of statistical method, scarce sensitivity.	[161,162,163]
MIP—Molecular Imprinted Polymer	Wheat	Low cost, stable, reusable.	Poor selectivity.	[164]
MIR—Mid Infra-Red	Cereals	Short or no sample preparation, greater absorption bands than NIR, high specificity, few overlaps.	Scarce low concentrations sensitivity/accuracy, in quantitative analysis non-comparable to NIR, insensitive to some substances.	[165]
Raman spectroscopy	Cereals	Narrow bands/few overlaps very high specific; good signal to noise ratio (SNR), short or no simple preparation, basically impervious to water, quick for complete analysis.	Weak raman effects: sensitive and higly improved instruments needed; inadequate for fluorescent samples, laser source can destroy samples, expensive experiment materials.	[166,167,168,169]
HSI and Multispectral VIS-NIRS	Cereals	Inexpensive, non-destructive sample, quick, grain sorting device wich classify individual contaminated grains; separation of safe from infected cereals.	Big data: enhanced hardware, data pretreatments and effective chemometric algorithms needed; expensive tools.	[170,171,172,173,174]
Quantitative NMR	Feed	Versatile technique with a broad range of applications: purity determination, metabolomic studies, multi-marker quantitation and quality control of samples.	Low sensitivity and spectral resolution, scarce time resolution, lack of selectivity/specificity limited quantification methods and lmited commercial software on the market.	[175,176]
Omic tools (proteomic, genomic, metabolomics, transcriptomics)	Cereals and others crops	Real time solution in pre and post-harvesting; identifying mycotoxins species; evidencing plant–fungal interactions; climate change impact on mycotoxin prevalence; provide information about early stage mycotoxin production biomarkers.	Under development, databases development, early studies.	[177]
Molecular techniques (PCR, FISH, DNA barcoding)	Wheat and maize	Development of mycotoxin genomes, better understanding the genes for biosynthetic production of mycotoxin.	More research needed for key genes of mycotoxin biosithesis and regulation factors including transcriptional regulation factors and environmental ones.	[178,179,180]
Electronic nose	Cereals	Rapid and powerful for quality control, management and research; assessment of physicochemical features of secondary fungal metabolites; detection of volatile compounds.	Future work on the sensor materials and data analysis and better understanding in industrial need related to quality control and monitoring of feed processing.	[181,182]
Electronic tongue	Cereals	Higly sensitive, selective, and low-cost method, it analize the liquids.	NF	[182]
Aggregation-Induced Emission Dye	Wine-coffee	Specificity for OTA recognition; very effective application: on-site food contaminations detection/simple operation.	NF	[183]
Antibodies	Food-Feed	Promising tool for the monitoring of food quality; involved in immunoassays.	Difficult to be expressed recombinantly and are susceptible to harsh environments.	[184]
Nano-bodies	Food	Expressed easily in prokariot and eukariot expression systems, strong in extreme terms and easy to use as replacers for artificial antigens.	Critical assessment of their performance.	[184]
Targeting peptides	Food	OTA peptide binding; reduce biosensor design cost, increase product life cycle, and semplify multi-analyte detection of mycotoxins.	NF	[185,186]
Emerging Sorting methods				
Optical sorting systems	Cereals	Detecting fungal contamination of seeds, indicating visible damage or discoloration, chemical constituents.	Performance varied depending on the seed, its orientation, or geometry, extent of infestation or contamination, and thresholds used.	[187]
Fluorescent spectroscopy	Cereals, oils	Quality and safety measurement for a variety of food and agricultural materials.	Strong dependence on autofluorescence of samples and on the intensity of incident light.	[188]

**Table 5 toxins-13-00572-t005:** Nanotechnologies in livestock production against mycotoxins.

Ano-Technology	Matrix/Animal	Results	References
Nano-biopolimers			
Yeast cell wall nano-biopolymer	Animal	Selectively bind adsorbable mycotoxins.	[257,258]
Chitosan polymeric nanoparticles	Feed raw materials; animals	Simultaneous adsorption of diverse mycotoxins.	[259]
Synthetic polymers			
Polystyrene nanoparticles	Animal feed	Polyethylene glycol linker and mannose targeting biomolecule bond and degraded toxigenic fungi.	[259]
Nano-clay binders			
Nanoclays	Animal feed	Aflatoxin adsorbing agents; can bind and detoxify Aflatoxin B;	[258,260]
Monmorillonite nanocomposite (MMN)	Poultry, food and feed products	Prevent aflatoxin toxicity and ameliorator of aflatoxicosis; adsorb ZEA in acqueos and organic solutions, commercially low cost product, high elimination effectiveness, flexible.	[261,262]
Nanosilicate platelets (NSP)	Feed	Adsorbing agents fumonisins B1; in vitro model.	[263]
Organimodified montmorillonites (OMNM)	Animal	Adsorbtion effects against AFB1 and OTA, finding it very safe and effective against different mycotoxins.	[264]
Carbon nano-materials			
Magnetic carbon nanocomposites	Poultry feed; animals	Set up from bagasse and alternative of granular adsorben carbon for reducing AFB1, efficient in the detoxification and degradation from gastrointenstinal segment of broiler chickens and exit no destructive effects.	[265]
Modified nanodiamonds	Animal intestine;	Mycotoxins intestinal adsorbent; provide the chemical structures (e.g., hydroxylation, carboxylation, and hydrogenation) to allow the surface functionalization, supplying binding affinity to different mycotoxins; mycotoxin adsorbtion (i.e., OTA and AFB1) rely on the functional groups of the nanodiamond.	[266,267,268]
Carbon nanotubes (CNTs)		single/multi-walled CNT applied for mycotoxin adsorption (trichothecenes, ZEN, and AFTs).	[128,268]
Magnetic graphene (MGO)	Palm kernel cake	MGO prepared by graphene oxide and iron oxide NPs, for mycotoxin adsorption.	[269]

## Data Availability

Not applicable.

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
