# Peer review of "Integrated Mycotoxin Management System in the Feed Supply Chain: Innovative Approaches"

_toxins, 2021, doi:10.3390/toxins13080572_

Round 1

Reviewer 1 Report

It's an useful summary of mycotoxin management program to reduce risks in the feed supply chain.

Final solutions are interesting and well argued.

The references are recent.

Author Response

Dear Reviewer,

Thank you for taking the time to read through our paper and thank you for your comments/feedback.

Reviewer 2 Report

Interesting paper and good materials, title: Integrated Mycotoxin Management System in the Feed Supply 2 Chain: Innovative Approaches.

Abstract:

The abstract needs to be re-written and rigorously developed in line with manuscript results. It is not clear about the originality of the study. It is not clear what an “updated integrated system” includes. You should provide a little bit more about the system. Please check the font as well as I think it is not consistent. It is not needed to put further study examples in the abstract.  It should be put in at the end of the paper. From the beginning until “bio and non…” has one font size, the rest has another font size.

Introduction:

Introduction part should re-written and reformulated more logically. The existing gap in the literature is not clear. Authors should emphasize on that. it also needs another para to talk about the remainder structure of the paper (last para).

Literature Review:

More comprehensive and critical review of literature is required. Currently, is more like descriptive analysis, while it should be more critical.

Feed supply chain is not clearly discussed except one figure. More clarifications about different stage of feed supply chain needs to be discussed.

The numbering in the literature review part is confusing, it can not be identified if we still within the lit review part or we moved to another section.

Figure 2 is not clear as well. authors should explain more clearly what is the purpose of using his figure and how different elements within that figure are related.

Solution Methodology:

No sign of methodology. It is not clear which methodology authors used to come up with those results. It should be developed more systematically.

Results and Discussion:

What is the novelty of the research works? What s the implications for mangers and also contributions for literature. It is not clear.

Conclusion:

Conclusion is not developed rigorously.  How about limitations and future research directions? It is expected to see these points in the conclusion section which is not clear in this current version.

Author Response

Dear Reviewer,

thanks for your comments and feedback.

Reviewer 3 Report

The article is representing a well documented and extensive description of mycotoxins occurrence and control throughout the logistic chain, from producers to market.

The references are up to date, the language is understandable and the arguments and discussion of findings coherent, balanced and compelling.

I recommend publication after  spell check of several minor issues  (ex CO2 instead of CO.....2 )

Author Response

Dear Reviewer,

thanks for your comments and feedback.

I have implemented the changes you suggested, I have found them very helpful and insightful.

Reviewer 4 Report

 This review article discusses innovative approaches to integrated mycotoxin management system in the feed supply chain.  Authors assert a need to study this topic due to exposure to mycotoxins being a worldwide concern for their ability to affect the quality and production of livestock and also act as carriers and pose a risk to human health.

The following feedback can be found below:

  1. Authors provide a compelling introduction as to why feed safety is an important precursor to food safety and why mycotoxin exposure should be reduced.
  2. The discussion on occurrence of mycotoxin is well-established in the paper.
  3. It is not immediately clear what the health effects of mycotoxin exposures are, particularly after it is stated that ” In a recent survey (Gruber-Dorninger et al., 2017), acute and chronic toxicity as well as occurrence data have been reported for most of the main emerging toxins. By far not 177 all of the detected compounds are toxicologically relevant at their naturally occurring levels and are therefore of little or no health concern to consumers”. (Lines 176-179). So, it that exposure to mycotoxin does not want any health concerns?
  4. Line 37 and throughout the document – It is not clear why the word ‘BIOMIN’ is in all capital letters?
  5. Line 41 – the acronyms WHO and FAO should be spelled out in entirety the first time they are introduced.
  6. Lines 80-81 – It is mentioned that mycotoxin risk in Europe is high to severe. What is this the case?  The authors should elaborate.
  7. Line 87- Check for extra space between two words in the sentence.
  8. The paper uses several acronyms which may not be as reader-friendly and perhaps not as palatable to a lay audience.
  9. Line 236 – For the Integrated Mycotoxin management system, who administers this system?
  10. Line 288 – In the table, who is responsible for harvest and storage? Both items have a blank under the ‘responsable’ column.
  11. Overall, this is an insightful, relevant, and interesting paper. Attending to some clarifying questions may help to improve the quality of the paper. 

Author Response

Dear Reviewer,

Thanks for your comments and feedback.

Round 2

Reviewer 2 Report

I am now happy with the current version.